# Buy and/or Pay Disparity: Evidence from Fully Autonomous Vehicles

Sunbin Yoo *, Junya Kumagai, Yuta Kawabata, Alexander Ryota Keeley and Shunsuke Managi

Department of Civil Engineering, Kyushu University, Fukuoka 819-0395, Japan;
j.kumagai.vj@adm.fukuoka-u.ac.jp (J.K.); syfgetg7@gmail.com (Y.K.);
keeley.ryota.alexander.416@m.kyushu-u.ac.jp (A.R.K.); managi@doc.kyushu-u.ac.jp (S.M.)
* Correspondence: yoo.sunbin.277@m.kyushu-u.ac.jp

**Abstract:** This study investigates the impact of environmental concerns, concerns about potential accidents, and the perceived advantages of fully autonomous vehicles on individuals' willingness to buy and the perceived value of these vehicles. Our research, conducted through a comprehensive survey with over 180,000 respondents in Japan and analyzed using structural equation modeling, reveals a nuanced disparity between willingness to buy and perceived value. We find that individuals concerned with the conservation of the natural environment are more likely to purchase fully autonomous vehicles due to their broader interest in societal issues and belief in the potential of new technologies like fully autonomous vehicles as solutions. However, these individuals attribute a lower perceived value to these vehicles, mainly because their adoption does not directly contribute to the conservation of the natural environment. Additionally, our results show that those recognizing the potential advantages of fully autonomous vehicle technology have a higher willingness to buy and perceived value, while those with apprehensions about the technology are less likely to purchase and attribute a lower perceived value to these vehicles. This study offers vital insights for policy and planning, highlighting the complex interplay of factors influencing the willingness to buy and perceived value of fully autonomous vehicles, critical for strategizing their adoption.

**Keywords:** fully autonomous vehicle; PV; structural equation model; environmental concerns



## 1. Introduction

The rapid advance of autonomous vehicles (AVs) and fully autonomous vehicles (FAVs) implies the arrival of the era of completely driverless cars. Starting from Google's self-driving car project in 2009, famous companies such as Uber, Apple, and Tesla have been challenging the development of autonomous vehicles. This is because shifting to FAVs from vehicles without any automation function would have numerous benefits if properly used, including the prevention of accidents due to human errors [1,2], alleviation of congestion [3,4], and reduction of emissions from traffic [5–7].

Due to these benefits, the market for fully automated vehicles begin to increase and the market experts predict the market share of FAV technology worldwide to reach 15–20% by 2025 [8], calling for the policies and future blueprints for the gradual shift to FAV. Drafting such standards would require understanding which factors would encourage or discourage potential consumers from adopting autonomous vehicles and, at the same time, how they evaluate them in monetary terms. Such an understanding requires including people's intentions and behaviors, determined by attitude and perceptions, which requires constructing latent factors [9]. Furthermore, people can have multiple attitudes simultaneously; for example, people can fear FAVs but at the same time appreciate the merits of FAVs. Thus, given that people simultaneously perceive benefits and fears from FAVs, s/he (as an inclusive, gender-neutral pronoun, intended to inclusively represent "she or he") may express high levels of WTB and PV because s/he appreciates the benefits more than s/he fears FAVs. Furthermore, such different attitudes might be correlated.

In this study, we constructed four categories of latents, considering their correlations, that express the attitudes and behaviors of people through an extensive literature review and estimated their relationship to WTB and PV. We also considered socioeconomic factors, such as income, gender, household size, and car-related factors, such as car ownership and car type. We chose structural equation modeling (SEM), a widely known methodology, to scrutinize people's psychometric intentions, which allows for the identification of latent factors and simultaneous estimation of latents with exogenous variables. Figure 1 shows our study structure. First, we identified attitudes, which are expressed in latents, according to the behaviors, and estimated the relationship between intentions and decisions.

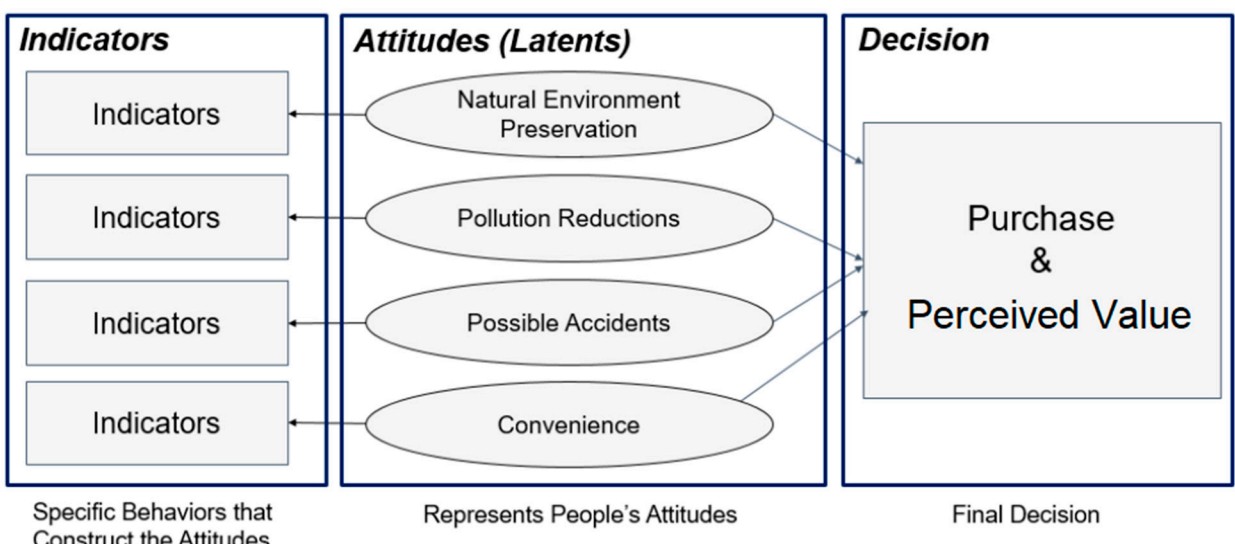

**Figure 1.** Study Structure.

Data-wise, we conducted a survey of more than 180,000 respondents in Japan, which contained questions regarding WTB and PV of FAVs, individual characteristics such as income, gender, commuting time and ages, environmental awareness, and opinions on the advantages and concerns regarding FAVs. Then, we constructed four empirical models that answer our research questions. Our results indicate that while environmental concerns enhance the likelihood of purchasing FAVs, they simultaneously diminish their perceived value due to the perception that FAV adoption does not directly benefit environmental conservation. On the other hand, individuals who recognize the benefits of FAV technology demonstrate a higher willingness to buy and assign greater value to these vehicles. Conversely, apprehensions regarding FAV technology correlate with a lower inclination to purchase and a reduced perceived value. These insights are pivotal in informing the development of effective policies and strategies for the adoption of FAVs.

The remainder of this paper is structured as follows. Section 2 provides a background in terms of industry and policy. The data and model are presented in Section 3. Section 4 shows the empirical results. Section 5 discusses our findings and provides policy implications. Section 6 concludes our paper.

## 2. Backgrounds and Literature Review

In this section, we first demonstrate the industry backgrounds in Section 2.1 and summarize previous works on FAVs in diverse aspects in Section 2.2.

### 2.1. Literature Review

#### 2.1.1. Expected Benefits and Concerns Regarding FAVs

The existing literature underscores the significance of examining the demographics of those adopting FAVs with a focus on the potential advantages and drawbacks of AVs or FAVs. This body of work posits that the anticipated benefits of AVs and FAVs can only be

actualized through user acceptance and proper utilization. This entails avoiding pitfalls such as neglecting improvements in fuel efficiency or rapidly increasing travel distances due to overreliance on these vehicles, thereby highlighting the pivotal role of consumers. Informed by these insights, our research question is oriented towards exploring consumer behavior patterns, specifically in terms of choice and PV, in relation to AV/FAV adoption.

FAV technologies are expected to change the transportation paradigm with minimized human interventions [10]. Benefits of FAVs include an increased/optimized traffic capacity [11,12] and reduced vehicle emissions [7,13]. Due to these advantages, AV technology has experienced explosive growth, and substantial recent literature focuses on the potential changes after introducing FAVs and AVs. These works include changes in travel behavior [14–19], and some of the works focus on changes in work–home location [20,21]. Some works focus on travel time savings [22–24], while the other strands of works look at the time-saving effects and merits of using AVs with public transportation choices [25–27]. Some of the studies look at the environmental benefits of using FAVs [6,28,29].

On the other hand, there are also several works highlighting the potential negative consequences of employing AVs; Refs. [30,31] state that policies need to intervene to reduce emissions from AVs, and [32] emphasizes that accidents may increase due to excessive trust in AVs. Ref. [33] show that autonomous vehicles have trouble reacting to the complex pedestrian environment. Thus, FAV drivers need to pay additional attention to protect pedestrian safety.

### 2.1.2. Hindrances Regarding FAV Choices

Then, why has FAV's market share been so low until now? Numerous studies show that the acceptability of autonomous vehicles is hindered because people are wary of various issues. First, previous works argue that people are reluctant to purchase FAVs because they fear potential traffic accidents. Fagnant and Kockelman (2015) mention that concerns regarding accountability/responsibility in traffic accidents can be an obstacle for potential FAV consumers [34]. Li et al. (2019) and Raj et al. (2020) point out that resolving the problems regarding the responsibility for damages is crucial for promoting FAV use [35,36]. These problems eventually discourage people from choosing FAVs [33,36]. Therefore, Morita and Managi (2020) mention that to promote use, credibility should be guaranteed [37]. Concerns about traffic accidents can also be extended to problems that might discourage technological improvements in FAV suppliers. Bansal and Kockelman (2017) mention that the appropriate regulations on safety norms can accelerate FAV technology innovations [38]. Bansal and Kockelman (2017) and Shladover and Nowakowski (2019) show that the absence of clearly defined safety norms would be a challenge for consumers to accept FAVs [38,39].

Second, the other strands of literature argue that people are not fully aware of the potential but substantial environmental benefits of FAVs; therefore, substitution toward FAVs is hindered. On the one hand, Bansal and Kockelman (2017), Shladover and Nowakowski (2019), Acheampong and Cugurullo (2019) and Haboucha et al. (2017) mention that pro-environmental consumers accept technological innovations if they can reduce pollution [38–41]. Similarly, Krueger et al. (2016) show that pro-environmental consumers are likely to choose FAVs [42]. On the other hand, Gkartzonikas and Gkritza (2019) show that consumers' lack of understanding of the environmental benefits of FAVs can be a barrier to FAV acceptance [43].

Thirdly, several studies have investigated the accident-related dimensions of autonomous vehicles. Notably, Morita and Managi (2020), as well as Yoo et al. (2023), have explored the decision-making processes of drivers and passengers in trolley-car dilemma scenarios. Specifically, Yoo et al. (2023) highlighted that potential buyers might be deterred from acquiring autonomous vehicles due to apprehensions about unavoidable accidents [37,44].

### 2.1.3. FAV Choices and Demands

The relatively early literature in this area examined purchasing decisions (or a choice) by analyzing the relationship between sociodemographic variables (i.e., gender, income, and age) and technological features in AV adoption. For example, some of the works look at the technological benefits (i.e., automatic braking and parking assistance) as the primary driver of AV purchases [45–47]. More recent literature looks at psychological aspects such as fear of the new technology [34,36,48,49]. Some of the works also look at the factors correlated to the PV of AVs [50–52], and most of them focused on the relationship between socioeconomic factors (i.e., income) and PVs.

Earlier studies mostly find that higher environmental concerns, higher income, and technological benefits are indeed the main drivers of AV and FAV adoption and higher PV. We, however, find some research gaps from these works; they primarily focus on either PV and WTB, and less attention is given to the factors affecting both, leaving out a potential distortion in the results due to the possibility that one might not have higher PV but has higher WTB, and vice versa. Consequently, these studies tend to imply that factors with higher PVs of FAVs would encourage FAV purchases. Such a conclusion may change, however, if the factors affecting WTB and PV are different. In fact, some of the previous works have already shown that the WTB and PV do not always align. Other than this research gap, we find some issues that our model can address, and we discuss all of them in Section 2.2.

### 2.2. Current Research Gaps and Our Contributions

This study aims to better understand the factors affecting the WTB and PV of FAV choices. This study has several contributions. First and most importantly, to the best of our knowledge, our model is the first to account for the differences between WTB and PV in the context of FAVs. Having a higher PV does not necessarily indicate that someone will purchase an FAV. For example, as mentioned in Section 1, one might evaluate the autonomous vehicle higher than others because s/he is aware of the benefits of FAV technology but would not purchase an FAV because s/he is afraid of potential accidents. Our work and results allow us to understand and distinguish the characteristics of individuals who belong to these groups (those who evaluate FAVs higher but do not purchase, and vice versa), and those who do not belong to this group are crucial to policymakers who are willing to spread the use of FAVs and bring them into the mainstream.

Second, we account for technological advantages, fears toward new technology, environmental awareness, and sociodemographic factors (i.e., gender, age, income, and commuting time) together in the model. Each factor encompasses crucial issues (i.e., cybersecurity, regulations on accidents, responsibilities on accidents, safety, and malfunction) closely related to the WTB and PV demands and carefully selected in the literature. One advantage of this approach is that we can separately analyze each factor's impact while fixing the other factor: for example, one might be afraid of FAV technology, but at the same time, s/he can support the conservation of the natural environment. Another example would be a person who fears FAV due to possible accidents or malfunction issues, but at the same time, s/he is fond of the advantages that FAV would give. In this case, there is a need to analyze these factors as independent factors separately. For example, our study allows us to look at the impact of fear toward accidents while fixing interest on the natural environment, therefore focusing on the changes of one while leaving the other as it is, and vice versa. Such approaches would allow us to evaluate each factor's 'independent' impact. Additionally, our model allows us to look at the impacts of each survey question on the factors. We discuss the factors more in Section 3.2.

Third, we categorize environmental concerns into two categories and investigate their correlations to the WTB and PV of FAVs. There are only a handful of studies on the impact of environmental concerns on FAV adoption and PV [53]. In a broader view, there are previous works on consumers' environmental concerns and their purchasing behaviors, and most of them agree that higher environmental concerns would lead to higher consumer preferences

for pro-environmental products [54–56]. Hence, one issue from these works is that they regard environmental awareness as a broad concept: public awareness of the importance of environmental protection. However, purchasing behavior might vary according to the type of environmental awareness. For example, the environmental benefits of adopting FAVs are mostly related to pollution and are not directly correlated to the conservation of the natural environment (i.e., biodiversity preservation). In that sense, those who prioritize these issues more than reducing air pollutants might not show higher PVs than those who regard resolving air pollution as important. Therefore, policy guidelines that do not consider differences between different environmental awareness types might result in misleading policy implications. To resolve this research gap, we categorize environmental awareness into 'Pollution', which refers to the people who emphasize recycling, alleviating air/water pollution, and 'Nature', which refers to the people who prioritize natural environments. Categorizing would also allow for the identification of which types of environmental awareness would be positively/negatively correlated with WTB or PV, respectively.

Hence, fourth, while our study looks at each factor's effects, we also allow the correlations between the different factors. Allowing such correlations is crucial because the WTB and PV choices would be influenced by multiple combinations of different factors. To be more specific, our model allows us to look at the 'independent' impacts of each factor. At the same time, our model would take into account the correlations of each factor in estimation. If we exclude one of each of the factors given that the factors may be correlated to each other, there will potentially be an omitted variable bias and endogeneity can occur. From the perspective of econometrics, ignoring such endogeneity can lead to incorrect estimation and might result in misleading policy implications.

## 3. Methodology

### 3.1. Data

In response to our research question, we conducted a comprehensive online survey in Japan from 16 November to 14 December 2015, targeting a diverse demographic to gather insights on fully automated vehicles (FAVs), lifestyle, and environmental concerns. The survey, executed through Nikkei Research Inc., was designed to randomly select participants while mirroring the gender and age distribution of the Japanese population. This approach ensured a representative sample, as evidenced by the 246,642 respondents who were rigorously screened through trap questions to maintain data quality.

To ensure the robustness of our findings, we employed multiple data processing methods. These included comparative analyses between the survey results and Japanese Census data, specifically examining the distribution of socioeconomic variables as shown in Table A5. Although we observed minor discrepancies in gender and education levels when compared to the census data, our survey results broadly align with Japan's average demographic trends. This methodology not only underscores the comprehensiveness of our data but also demonstrates our commitment to utilizing varied analytical approaches to validate our findings.

Again, we are aware that in 2015, the respondents were less familiar with FAVs than those in 2021. Therefore, we excluded those who answered that they had 'no awareness' of FAVs, which accounted for 14.48% of the entire sample (35,715 observations). Thus, in our model, we only account for the people who were aware of FAV technologies in 2015. Therefore, given that FAVs were not introduced in the market back then and still not introduced in 2021, a substantial change in the result, for example, a change in the sign or implications of the results, is less likely to happen. Therefore, more attention should be given to the signs and relative comparisons of coefficient magnitudes of the latent constructs.

Finally, we drop those who selected "don't know/don't want to answer" about their individual income (30,156 observations). As a result, we have 180,771 respondents in total. Before the large-scale survey started, a pre-survey was carried out to tune the questionnaires.

For the questions related to the FAV purchasing intention, respondents were asked the question: "Do you want to add a completely self-driving option that allows you to move around when you purchase a car in the future?". Then, the respondents answered the following questions: "(1) Purchase for sure, (2) Purchase under certain conditions, (3) Do not purchase, and (4) I don't know". Given that FAVs are not yet fully introduced to the market in 2015 or 2021, we assume that people who show an affinity to FAVs can be potential consumers in the future. Therefore, we included those who answered (1) and (2) as a group of 'potential consumers' as they show an affinity toward using FAVs. On the other hand, people who answered (3) and (4) are reluctant to purchase FAVs, and we did not consider them potential consumers. Therefore, we coded WTB equal to 1 if a respondent belonged to a potential consumer group and coded WTB as 0 if not. Therefore, our analysis would allow us to see what kinds of factors would shift consumers who belong to (3) and (4) to (1) and (2). We would like to note that we are making a clear distinction between "adding" a completely self-driving option and "purchasing" FAVs by asking "Do you want to add a completely self-driving option that allows you to move around when you purchase a car in the future?".

Next, we also asked PV for FAVs. Respondents were asked to write down their PVs freely regardless of the purchase decisions, ranging from 0 JPY to 3.25 million JPY (based on the approximate conversion rate of 1 JPY to 0.0073 USD, the range of PV for FAVs from 0 JPY to 3.25 million JPY, converts to approximately 0 to 23,725 USD). We used a payment card method to measure PV, and we provided detailed ranges of PVs in Table 1. However, given that FAVs are a newly introduced technology, people may not have a specific price range of PV if we choose to leave PV as an open question. In that sense, leaving PV as an open question may increase the variances of responses for two reasons. First, because evaluating PV is not a typical daily decision-making behavior, it may result in many nonresponses, and respondents would feel that it is difficult to answer with a concrete number without providing any examples. Second, following the first reason, the number of outliers may increase, and the outliers may distort the representative values by abnormally large or small amounts. Third, the answers tend to be concentrated on round numbers (Ministry of Land, Transport, and Infrastructure—We refer to https://www.mlit.go.jp/kowan/beneki/images/kaigan_hiyoubeneki_06.pdf, accessed on 26 October 2023). Thus, we chose to use categorical but detailed PV questions. We have respondents who chose a PV of 0, indicating that they would choose to add it if it is free, and such an answer does not indicate that the respondents are not willing to purchase AVs.

**Table 1.** The range, frequency and respondents' proportion of the willingness-to-pay (PV) in our survey.

| Please Write Down Your Willingness-to-Pay for Adding Fully Automated Option to Your Newly Purchased Vehicle. | | | |
|:---:|:---:|:---:|:---:|
| **Group** | **Range (10,000 JPY)** | **Frequency** | **Proportion (%)** |
| 1 | 0 | 40,093 | 22.18 |
| 2 | 1–5 | 34,666 | 19.18 |
| 3 | 6–10 | 27,456 | 15.19 |
| 4 | 11–15 | 11,331 | 6.27 |
| 5 | 16–20 | 14,987 | 8.29 |
| 6 | 21–25 | 4067 | 2.25 |
| 7 | 26–30 | 18,511 | 10.24 |
| 8 | 31–35 | 2346 | 1.30 |
| 9 | 36–40 | 1179 | 0.65 |
| 10 | 41–45 | 569 | 0.31 |
| 11 | 46–50 | 9524 | 5.27 |

**Table 1.** *Cont.*

| | Please Write Down Your Willingness-to-Pay for Adding Fully Automated Option to Your Newly Purchased Vehicle. | | |
|---|---|---|---|
| Group | Range (10,000 JPY) | Frequency | Proportion (%) |
| 12 | 51–60 | 1549 | 0.86 |
| 13 | 61–70 | 399 | 0.22 |
| 14 | 71–80 | 1868 | 1.03 |
| 15 | 81–90 | 1261 | 0.70 |
| 16 | 91–100 | 5610 | 3.10 |
| 17 | 101–150 | 1441 | 0.80 |
| 18 | 151–200 | 1053 | 0.58 |
| 19 | 201–250 | 479 | 0.26 |
| 20 | 251–300 | 465 | 0.26 |
| 21 | 300+ | 1917 | 1.06 |

We also included the respondents' car ownership and car types in our model for two reasons. First, we would like to increase the survey's internal validity; therefore, we would like to control for individuals who do not know the price and maintenance costs for cars. Thus, we included the 'car ownership' variable to control for those who do not own a car and are less likely to be aware of car prices. Second, along with car ownership, we also include car types (gasoline, diesel, hybrid, plug-in-hybrids (PHEV), fuel-cell vehicle (FCV), and electric vehicles (EV)), as car prices differ according to the car types.

Then, we asked about concern for the environment in the form of 'importance as a policy' on a 5-point Likert scale, including zero (no awareness). Based on previous studies, we classified the topics for environmental policy into eight factors referring to the House of Councilors, The National Diet of Japan, (2015) [57]; We have 13 questions in total, and the topics are about the renewable energies, air pollution, environmental conservation, water pollution, endangered species conservation (biodiversity), reuse and recycling, waste disposal, and CO2 emissions with questions such as, "How important is the policy to you?' The scale of responses is as follows: (0) for no awareness/interest at all--therefore, the difference between those who answer (0) and others would be whether that person at least has an interest in a certain policy/issue, (1) for very insignificant; (2) for insignificant; (3) for neither important nor insignificant; (4) for important; (5) for very important. Next, we surveyed the technological merits and concerns regarding FAVs. Respondents were asked to check multiple options among 17 options for merit and 12 options for concerns.

We also included sociodemographic variables: income, gender, age, and commuting time. Table 2 shows descriptive statistics. Overall, we had approximately 180,771 respondents. We divided the sample into three groups: the overall group (Panel (A)), those who would not purchase an FAV, (as in Panel (B)), and those who would purchase an FAV (as in Panel (C)). Although we do not see significant differences across the groups for the sociodemographic variables, annual income, PV for FAVs, and EV dummy show higher mean values for those who belong to Panel (C) than in Panel (A) and (B).

**Table 2.** Descriptive statistics.

| Variable | Mean | Std.dv | Min | Max |
|---|---|---|---|---|
| Panel (A) Overall (N = 180,771) | | | | |
| PV for FAV (10,000 JPY) | 22.519 | 44.275 | 0 | 325 |
| Annual Income (10,000 JPY) | 485.383 | 411.226 | 100 | 3500 |
| Household Size | 2.864 | 1.362 | 1 | 10 |
| Age | 48.701 | 11.933 | 18 | 100 |
| Female Dummy (=1 if female) | 0.369 | 0.482 | 0 | 1 |
| Married Dummy (=1 if married) | 0.695 | 0.461 | 0 | 1 |
| Car Ownership (=1 if own car) | 0.823 | 0.381 | 0 | 1 |
| Gasoline (=1 if car type is gasoline vehicle) | 0.676 | 0.468 | 0 | 1 |
| Diesel (=1 if car type is diesel vehicle) | 0.023 | 0.150 | 0 | 1 |
| Hybrid (=1 if car type is hybrid vehicle) | 0.116 | 0.321 | 0 | 1 |
| Plug-in Hybrid (=1 if car type is plug-in hybrid vehicles) | 0.004 | 0.065 | 0 | 1 |
| EV (=1 if car type is electric vehicles) | 0.002 | 0.049 | 0 | 1 |
| FCV (=1 if car type is fuel cell vehicles) | 0.0005 | 0.022 | 0 | 1 |
| Panel (B) People who will not choose autonomous vehicles (N = 77,371) | | | | |
| PV for FAV (10,000 JPY) | 19.449 | 46.026 | 0 | 325 |
| Annual Income (10,000 JPY) | 446.782 | 391.013 | 100 | 3500 |
| Household Size | 2.833 | 1.351 | 1 | 10 |
| Age | 48.833 | 11.980 | 18 | 100 |
| Female Dummy (=1 if female) | 0.415 | 0.493 | 0 | 1 |
| Married Dummy (=1 if married) | 0.685 | 0.465 | 0 | 1 |
| Car Ownership (=1 if own car) | 0.821 | 0.383 | 0 | 1 |
| Gasoline (=1 if car type is gasoline vehicle) | 0.693 | 0.461 | 0 | 1 |
| Diesel (=1 if car type is diesel vehicle) | 0.022 | 0.145 | 0 | 1 |
| Hybrid (=1 if car type is hybrid vehicle) | 0.101 | 0.301 | 0 | 1 |
| Plug-in Hybrid (=1 if car type is plug-in hybrid vehicles) | 0.003 | 0.056 | 0 | 1 |
| EV (=1 if car type is electric vehicles) | 0.002 | 0.045 | 0 | 1 |
| FCV (=1 if car type is fuel cell vehicles) | 0.0006 | 0.024 | 0 | 1 |
| Panel (C) People who would choose autonomous vehicles (N = 103,400) | | | | |
| PV for FAV (10,000 JPY) | 24.817 | 42.775 | 0 | 325 |
| Annual Income (10,000 JPY) | 514.266 | 423.430 | 100 | 3500 |
| Household Size | 2.887 | 1.370 | 1 | 10 |
| Age | 48.602 | 11.897 | 18 | 100 |
| Female Dummy (=1 if female) | 0.334 | 0.472 | 0 | 1 |
| Married Dummy (=1 if married) | 0.702 | 0.457 | 0 | 1 |
| Car Ownership (=1 if own car) | 0.825 | 0.380 | 0 | 1 |
| Gasoline (=1 if car type is gasoline vehicle) | 0.664 | 0.472 | 0 | 1 |
| Diesel (=1 if car type is diesel vehicle) | 0.024 | 0.153 | 0 | 1 |
| Hybrid (=1 if car type is hybrid vehicle) | 0.128 | 0.334 | 0 | 1 |
| Plug-in Hybrid (=1 if car type is plug-in hybrid vehicles) | 0.005 | 0.072 | 0 | 1 |
| EV (=1 if car type is electric vehicles) | 0.003 | 0.052 | 0 | 1 |
| FCV (=1 if car type is fuel cell vehicles) | 0.0004 | 0.020 | 0 | 1 |

Among all options and questions, we used factor analysis to choose the options that are used in the estimation. We discuss more on factor analysis and how we chose the important factors in Section 3.2. Specific lists of questions are listed in Table 3, which shows notations for each option and explanations of them. 'Sources' in Table 3 refers to the previous works we referred to when designing survey questions. The proportions of consumers choosing each option are listed in Appendix A, Tables A1 and A2.

**Table 3.** The List of Latent Variables.

| Notation | Explanation | Source |
|---|---|---|
| | *Fear (FE)* | |
| FE1 | There is a possibility that children will be able to move on their own. | [40,58–64] |
| FE2 | There is a possibility that the software is hacked. (Cyber security) | |
| FE3 | The malfunction may cause accidents. | |
| FE4 | It is unclear who is responsible for the accident due to FAV technology. | |
| FE5 | Traffic volume and congestion might increase as those without a license can drive. | |
| FE6 | The malfunction may lead to wrong destinations. | |
| | *Merits (MR)* | |
| MR1 | People can drive without a license. | [59,65–68] |
| MR2 | Burdens on driving would be decreased. | |
| MR3 | Children can move without a guardian. | |
| MR4 | Able to do other work while driving. (Multitask) | |
| MR5 | Able to avoid responsibilities of traffic accidents. | |
| | *Pollutants (EP)* | |
| EP1 | Recycling is important. | [69–73] |
| EP2 | Cycle utilization rate: the percentage of the total amount of reusable and recycled materials to be injected into society, is important for preventing pollution. | |
| EP3 | I think water quality should be improved. | |
| EP4 | Alleviating Particulate Matter (PM) 2.5. pollution is critical for our society. | |
| EP5 | Resolving air pollution (particularly, photochemical smog) is important. | |
| | *Nature (EN)* | |
| EN1 | Preserving endangered species is important. | [53,66,74–76] |
| EN2 | Preserving living animals (overall) is important. | |
| EN3 | The ratio of green area within 1500 m around a house is important. | |
| EN4 | Green purchasing: When purchasing goods and services, consider the environmental impact before purchasing. | |

### 3.2. Empirical Strategy

We use structural equation modeling (SEM) to assess the relationship between factors that are correlated with the WTB and PV of FAVs. We chose SEM, which is a suitable methodology that allows us to examine the psychometric factors that are correlated with people's intentions to FAVs. SEM can handle a substantial number of endogenous and exogenous variables and can include latent variables in the model. Thus, SEM enables the inclusion of the theory of planned behavior (TPB), which explains people's behavior based on psychometric intentions through latent variables determined by attitudes [77]. Thanks to such benefits, SEM has been employed in many research fields incorporating psychometric modeling, such as psychology, sociology, educational research, political science, and market research. Several SEM applications in transportation research have been conducted in the past (examples of previous works including SEM as the main method include [78–81]). Our model explains the WTB and PV of automated vehicles based on the four latents of nature, pollution, merit, and accidents and thus focuses on the psychometric intentions of the potential consumers, and SEM allows such analysis.

Moreover, SEM offers simultaneous estimations of latent variables and exogenous variables and allows for correlations between latents. If the latents and exogenous variables are estimated sequentially, for example, one can conduct factor analysis to construct the

latents in the first step and proceed to the estimation of latents and exogenous variables to the choice modeling, while this strategy is simple, it does not guarantee unbiased estimators for the parameters involved and tends to underestimate standard errors (see, for example, [82,83]). Furthermore, sequential estimation does not allow for the interaction of latent variables. As we assume that latents are correlated and people's choice behavior is not 'sequential', we choose SEM in this study and use STATA to estimate our model (see [84] for a discussion of sequential versus simultaneous estimation).

### 3.2.1. Identifying Latent Constructs

We first identified the latent variables that can be related to WTB and PV for FAV based on the process used by previous studies (e.g., [85]), as shown in Table 3. We chose four categories: fear (fear of FAV technology), merits (advantages and benefits of FAV technology), pollution (concerns about pollution), and nature (concerns about conserving natural environments) as the latent variables.

We conducted an extensive literature review and factor analysis to sufficiently validate our latent variable construction process. To do so, we focused on the merits of FAVs and focus on the disadvantages that FAVs would possibly bring. First, the latent variables and statements (questions) for each survey were based, whenever possible, on statements previously used and found to be effective in the literature. Second, we constructed the latent variables according to our research hypothesis, exploratory factor analysis (EFA) and previous works. First, using EFA, we explored the latent variables that represent the respondents' awareness and attitudes toward issues related to FAV and the natural environment—as a rotation method, we adopted the promax method, one of the oblique rotations, to assume that latent variables can be correlated with each other. In previous studies, orthogonal rotation methods are frequently used for setting no correlation between latents. However, it is debated that the uncorrelation assumption is unrealistic: in social science, attitudes and perceptions tend to be mutually related [86]. From the EFA, we obtained four latent variables: fear, merits, pollutions, and nature. These latent variables were derived from the indicator variables shown in Table 3. Cronbach's alpha values of merit, fear, pollution, and nature were 0.559, 0.734, 0.953, and 0.914, respectively. Cronbach's alpha is regarded as a measure of scale reliability, whose acceptable range is >0.6. Only merit did not satisfy this condition, but its Cronbach's alpha value was not too far from 0.6 [87]. The correlation between indicator variables is shown in Tables A3–A6 in Appendix A.

Next, using the relationship between latent and indicator variables obtained from the EFA, we conducted a confirmatory factor analysis (CFA) to estimate the coefficient of latents on indicators, and calculate the score of each latent variable. In the CFA process, we can assume the correlations between error terms of indicator variables. Suppose that one latent construct is measured by five indicator variables. The error terms of the indicator variables are calculated as their unique variance that is not related to the latent construct. If two specific indicator variables are similar compared to the other three, the two share common variances that are not captured by the latent. In such a situation, setting a correlation between the error terms of those two indicators can explain such a similarity and improve the overall model fit. We decided which error terms should be correlated with each other according to the goodness-of-fit indices and the strength of the correlations between indicator variables.

Finally, we included the four latent variables obtained by the EFA and CFA processes in our SEM model. These latent variables were used as the exploratory variables for purchasing decisions and PV for FAV. In addition, we included gender, individual income, age, and commute time as the control variables for purchasing decisions and PV for FAV because these individual characteristics may affect purchasing intention and PV as well as latent awareness and attitudes.

The first latent construct, fear, represents an individual's concerns toward possible accidents, malfunctions, or responsibility issues (i.e., who would be responsible when there is an accident) toward FAVs. Numerous works and experts argue that FAVs will eliminate

human errors, therefore creating safer traffic environments. Nevertheless, many members of the public are concerned about potential problems. These concerns were also mentioned in previous works; Petrovic et al. (2020) [61] mention that rear-end collisions are likely to occur more often in AVs. Ahmed et al. (2020) [62] argue that the public is still concerned about possible crashes due to malfunctions of AVs and cybersecurity issues. Other works also point out that people are concerned with safety issues [63]. Due to these concerns, we expect those who are wary of possible accidents to be less willing to purchase FAVs and AVs than those who do not fear them. On the other hand, resolving such issues would then encourage them to purchase FAVs and AVs [64].

The second latent construct, 'merit', shows an individual's interest in the advantages that AVs/FAVs would bring. It ranges from simple benefits that allow people without licenses or people without long-term experiences in driving to drive [68], to enable drivers to multitask [65], drive more comfortably [66], and usefulness [59,67].

The third and fourth latent variables are related to the environmental awareness of individuals. The third latent construct, 'pollution', represents attitudes about reducing environmental pollution and promoting reusing and recycling materials. The fourth is 'nature', which shows individuals' awareness about conserving biodiversity and the natural environment. Studies in the field of transportation show that an individual with high pro-environmental awareness has a higher intention to buy FAV [53,66]. Although most of the previous studies have only focused on overall pro-environmental attitudes, we categorized environmental awareness into pollution-related and conservation-related because each of them might have varied effects on attitudes toward AV. The contribution of AVs to the environment is associated with pollution reduction (particularly those related to air pollution) by easing traffic jams rather than the conservation of natural environments such as animals and forests. Thus, to promote AVs effectively, it is important to know whether both types of awareness, AV-related (pollution) and non-AV-related (nature), affect PV and WTB for AVs.

### 3.2.2. Structural Equation Modelling

Using the latent constructs, we have created SEM models as in Figure 2. The rectangles in the diagram symbolize the observed variables, whereas the circles denote latent variables and error terms. Each arrow represents the path from one variable to another, with bidirectional arrows indicating correlations between variables. Each latent variable is measured by its corresponding indicator variables. Subsequently, four latent variables and individual characteristics have paths to our primary objective variables: WTB and PV.

We have three models in total. First, we investigate factors that are correlated to WTB (Model (1)) and PV (Model (2)). Second, we assume that a higher PV would be positively correlated with a higher WTB; therefore, we add such a relationship to Model (1) and assume that all types of latents and other exogenous variables are correlated to both WTB and PV (Model (3)). Our preferred main model is Model (3), and we take Models (1) and (2) to confirm our findings in Model (3). Such diverse specifications from Models (1) and (2) allow us to confirm the robustness of the results. To make a better fit of the model, we assume that some of the error terms associated with indicator variables are correlated. Hypothesizing a correlation between these error terms can improve our model's ability to explain the data.

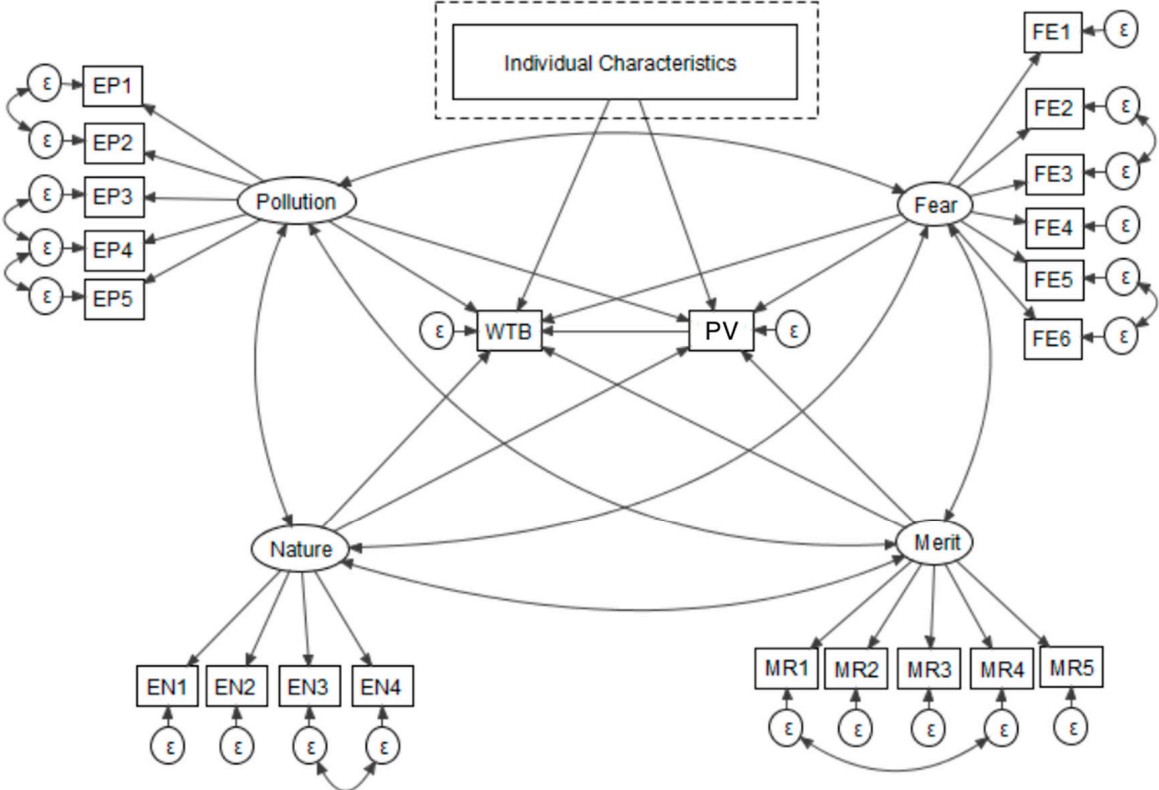

**Figure 2.** Conceptual framework.

## 4. Result

The results of the structural equation are shown in Table 4. In Table 4, the upper column shows the estimation results of PV, and the lower column shows the estimation results of WTB, of Models (1) to (3). Standardized coefficients are used to enable a comparison between the magnitudes of the coefficients. This type of coefficient displays the change in a dependent variable when an explanatory variable increases by one standard deviation. Thus, standardized coefficients are frequently used in quantitative studies as the relative importance of explanatory variables within a model [88]. While we have three models from Models (1) to (3), the estimated coefficients are similar across the models. Our models mainly show the WTB/PV disparity in regard to environmental concerns. The results of the measurement equation are shown in Table A4 in Appendix A.

### 4.1. Notes for Interpretations

We would like to clarify that people can have different combinations of latents. For example, people can have high levels of both 'Fear' and 'Merit', or lower levels of 'Merit' and 'Nature' and so on. Therefore, interpretations of our results should be made carefully. For example, it is concerns about accidents that are negatively correlated with PV, and it does not indicate that a person with high levels of 'Fear' does not appreciate the benefits of FAVs. Appreciation of the benefits from FAVs would be expressed in the coefficients of 'Merit'. Thus, it is possible to have both high levels of merit and fear. Our result shows the changes in WTB and PV after one unit of standard deviation increases in a latent state, keeping other latents fixed.

**Table 4.** Coefficient estimates of structural equation (N = 180,771). * $p < 0.1$, ** $p < 0.05$, *** $p < 0.01$. *p* stands for *p*-value. Standard errors in parentheses.

| Variables | Model (1) | Model (2) | Model (3) |
|---|---|---|---|
| | *Dependent Variable: PV* | | |
| **Latents** | | | |
| Nature | −0.029 *** 0.004) | −0.030 *** (0.004) | |
| Pollution | 0.086 *** (0.004) | 0.086 *** (0.004) | |
| Merit | 0.148 *** (0.003) | 0.152 *** (0.003) | |
| Fear | −0.021 *** (0.003) | −0.023 *** (0.003) | |
| **Socio-Economics** | | | |
| ln (Income) | 0.069 *** (0.003) | 0.069 *** (0.003) | |
| Household Size | −0.011 *** 0.003) | −0.011 *** (0.003) | |
| *ln (age)* | 0.026 *** (0.003) | 0.026 *** (0.003) | |
| Female | −0.001 (0.003) | −0.0008 (0.003) | |
| Marry | 0.022 *** (0.003) | 0.022 *** (0.003) | |
| **Car-Related** | | | |
| Car Ownership | −0.010 *** 0.025) | −0.010 *** (0.002) | |
| FCV | 0.008 *** (0.002) | 0.008 *** (0.002) | |
| Diesel | 0.011 *** (0.002) | 0.011 *** (0.002) | |
| Hybrid | 0.045 *** (0.002) | 0.045 *** (0.002) | |
| Plug-in Hybrid | 0.024 *** (0.002) | 0.024 *** (0.002) | |
| EV | 0.016 *** (0.002) | 0.016 *** (0.002) | |
| Constant | 0.550 *** (0.045) | 0.550 *** (0.045) | |
| | *Dependent Variable: WTB* | | |
| PV | | | 0.172 *** (0.002) |
| **Latents** | | | |
| Nature | 0.072 *** (0.004) | | 0.077 *** (0.004) |
| Pollution | 0.048 *** (0.004) | | 0.033 *** (0.004) |
| Merit | 0.240 *** (0.003) | | 0.215 *** (0.003) |
| Fear | −0.076 *** (0.003) | | −0.073 *** (0.003) |
| **Socio-Economics** | | | |
| ln (Income) | 0.076 *** (0.003) | | 0.064 *** (0.003) |
| Household Size | 0.006 ** (0.003) | | 0.008 *** (0.003) |
| ln (age) | −0.067 *** (0.003) | | −0.071 *** (0.003) |
| Female | −0.049 *** (0.003) | | −0.049 *** (0.003) |
| Marry | 0.013 *** (0.003) | | 0.009 *** (0.003) |
| **Car-Related** | | | |
| Car Ownership | −0.008 *** (0.002) | | −0.007 *** (0.002) |
| FCV | −0.006 *** (0.002) | | −0.007 *** (0.002) |
| Diesel | 0.007 *** (0.002) | | 0.005 ** (0.002) |
| Hybrid | 0.035 *** (0.002) | | 0.028 *** (0.002) |
| *Plug-in Hybrid* | 0.011 *** (0.002) | | 0.007 *** (0.002) |
| EV | 0.006 *** (0.002) | | 0.004 (0.002) |
| Constant | 1.618 *** (0.044) | | 1.524 *** (0.043) |
| **Fit Indices** | | | |
| RMSEA | 0.069 | 0.069 | 0.067 |
| AIC | $6.01 \times 10^6$ | $6.41 \times 10^6$ | $6.65 \times 10^6$ |
| CFI | 0.843 | 0.842 | 0.843 |
| GFI | 0.842 | 0.842 | 0.843 |
| AGFI | 0.802 | 0.802 | 0.81 |

### 4.2. WTB and PV

Throughout Models (1) to (3), we find positive correlations between WTB and PV of approximately 0.172, indicating the 'overall' trend that people with a high level of WTB are likely to have higher PV and vice versa. Nevertheless, whether individuals' attitudes, expressed in latent factors, also show positive (or negative) trends in both WTB and PV needs to be clarified. As mentioned in Section 3.2, if there is a disparity in WTB/PV in the latents, then the changes in latents might divert the overall relationship between WTB and PV. Furthermore, as people can have multiple latents, looking into how the

individual latents are correlated would also implicate which aspects and how much people are attracted/not attracted to FAVs. Implications from these results can also contribute to policies on motivating people to adopt FAVs. In this study, we find such a trend in the latents that are related to environmental concerns.

### 4.3. Environmental Concerns

Although 'Nature' and 'Pollution' are positively correlated with WTB, 'Nature' is negatively correlated with PV, while 'pollution' is positively correlated with PV. This result is interesting because it shows that environmental concerns can have different implications according to the types of concerns. Therefore, FAVs may be more attractive to people with higher levels of 'Pollution' than those who have higher levels of 'Nature'. We further discuss this result in Section 5.

### 4.4. Merits and Fear

As expected, merit shows positive coefficients in both WTB (0.215 in Model (3)) and PV (0.172 in Model (3)), while fear presents negative coefficients in both WTB ($-0.073$ in Model (3)) and PV ($-0.023$ in Model (3)). Such results are natural in the sense that people who appreciate the benefits of using FAVs would have higher WTB and PV, and those who fear potential accidents would not be more likely to purchase FAVs and would not appreciate FAVs more than those who do not fear FAVs.

### 4.5. Other Variables

Other socioeconomic variables, such as the income and 'marry' dummy variables, show positive coefficients for both WTB and PV, while household size, age, and gender show mixed conclusions, showing that factors affecting WTB and PV are different according to socioeconomic group. Car ownership shows negative coefficients toward WTB and PV, and this result implies that those who own and drive a car are unlikely to show high WTB and PV compared to those who do not own a car. Taking gasoline cars as a baseline, hybrid car owners would show the highest WTB and PV compared to other car types.

### 4.6. Model Fit

According to the goodness-of-fit indices shown at the bottom of Table 4, in general, the models fit the data modestly well. The acceptable range of RMSEA is <0.08, and those of CFI, GFI, and AGFI are <0.90 [87,89]. In our model, the values of RMSEA, CFI, GFI, and AGFI are generally within or near each variable's acceptable range.

Table 5 shows the correlation among the predicted scores of the four latent variables. Pollution and nature are strongly correlated, which implies that people who are concerned about a reduction in environmental pollution are also interested in the conservation of the natural environment. Merit and fear are also moderately correlated, meaning that people feeling merit from self-driving technology are also worried or scared about possible malfunctions and the negative influence of FAVs. Other combinations of latent variables are also correlated with each other, but the magnitudes are relatively smaller.

**Table 5.** Correlation matrix among the predicted scores of latent variables.

|  | **Pollution** | **Nature** | **Merit** | **Fear** |
|---|---|---|---|---|
| Pollution | 1 |  |  |  |
| Nature | 0.7990 | 1 |  |  |
| Merit | 0.2245 | 0.1823 | 1 |  |
| Fear | 0.3413 | 0.2122 | 0.4514 | 1 |

## 5. Discussion

In this section, we provide the implications of our results in Section 4, referring to previous works. We first provide a discussion of our results in Section 5.1 and policy implications for adopting FAVs in Section 5.2.

### 5.1. Overall Discussion

FAV technology is a newly introduced technology that needs broader public acceptance. Thus, it would face approval and disapproval from the public. People can respond to such technology by considering purchasing it as consumers (in the form of WTB) or show higher monetary appreciation (in the form of PV). If factors correlated with WTB are not linked or offer different implications to PV, then WTB and the PV of FAVs should be separately examined. To this end, this study validates it using structural equation models. Our findings allow us to answer the questions suggested in Section 1:

Our results indicate that environmental awareness, the advantages of using FAVs, fear of potential accidents, and socioeconomic factors are correlated with the WTB and PV of FAVs. We found a form of WTA-PV disparity. Those who are highly interested in the conservation of the natural environment may purchase FAVs but would show lower PV than those who are not interested. On the other hand, those who are interested in pollution alleviations show high WTB and PV.

Then, why does WTB-PV disparity occur? One may think it is natural to have higher PV if a person has higher WTB and vice versa. However, previous works in marketing, finance and economics state that WTB is positively correlated with social externalities [90], particularly for environmentally conscious consumers. This is because they perceive their social roles in which they are acting [91]. These people are usually interested in resolving overall social problems [92–95]. Particularly in the field of transportation, environmentally cautious consumers would be early adopters of new technologies (such as autonomous vehicles and battery electric vehicles) that may improve society even if the new technology is not yet mature [96–98]. Those who perceive themselves as environmentally conscious, therefore, may choose to purchase if they regard FAV technology as beneficial for the overall society by reducing emissions and congestion and providing convenient services. Thus, those who are interested in the conservation of the natural environment may show higher WTB.

What kinds of factors are correlated with higher PV? Previous works [99–102] answer this question by showing that PV is more closely correlated to 'private (or financial) benefits' rather than 'social benefits'. In other words, consumers may not show higher PV if the benefits from FAVs are not directly correlated to their private interests regardless of the benefits that society would receive. In sum, WTB is positively correlated with the advantages that society overall acquires, and this relationship may be powerful to those who are environmentally cautious. On the other hand, PV is correlated to the direct and private benefits that the individual would receive, as mentioned above.

In that sense, we can explain our results as follows: (i) those who are highly interested in preserving the natural environment may not appreciate FAVs; they would show lower PV. This is because FAVs do not have direct advantages that contribute to preserving the natural environment. For example, increasing the market share of FAVs would not conserve biodiversity or increase the size of green parks. Meanwhile, those interested in conserving the natural environment are interested in overall social issues and believe adopting FAVs would better society; these people would purchase FAVs and therefore would show higher WTB. (ii) Those who fear FAVs are fearful of accidents; they would be reluctant to purchase them, and such concern would be negatively correlated with PV.

By recognizing the form of the WTB–PV disparity, our result suggests that governments and industries may take an additional look at consumers with such disparities. Doing so could continue to expand the market share of FAVs in the future.

### 5.2. Implications on Future Adoptions of Autonomous Vehicles

FAVs would provide numerous social benefits. To realize those benefits and to accelerate the market introduction of FAVs, it would be necessary to increase consumer acceptance and evaluation. In that sense, our results suggest important implications for future adoptions of FAVs. First, people are still wary of potential accidents and malfunctions. Technological innovations can alleviate these concerns, and by doing so, WTB can increase. Next, for environmentally conscious consumers, conducting further assessments and appealing to the potential benefits of reducing energy use and pollution increase their PV toward FAVs. Our results show that the correlation between the latent constructs "nature" and "pollution" is highly correlated (79.90%), which implies that people interested in alleviating pollution are also interested in the conservation of the natural environment. Increasing PV by appealing to the potential benefits of reducing energy use and pollution to people who belong to "pollution" would positively correlate with the overall rise of PV in environmentally conscious consumers—of course, another way to increase PV is to promote that FAVs can also 'directly' contribute to the conservation of the natural environment, which needs a careful approach. In the long-term, the proliferation of FAVs may reduce travel distances and fuel consumption by reducing congestion, therefore green parks might increase in the future. However, as such benefits have not yet been well-examined until now, we decided not to include them in our main policy implications.

One concern from previous works and industry reports was that FAVs might increase the total vehicle miles traveled (VMT), as they allow many people to travel freely [103]. The impact of FAVs on energy use and emissions would largely depend on their effect on the total VMT and their fuel efficiency and fossil fuel consumption. For example, Stephens et al. (2016) [104] estimate that with the largest total vehicle travel increase and the smallest efficiency increase could result in as much as a 205 percent increase in US transportation energy use. On the other hand, the FAVs would have the smallest increase in total VMT with the largest efficiency increase, and FAVs can result in a 58 percent drop in energy use. Another way to reduce emissions and energy consumption is to promote shared FAVs, which can add up to 10% more travel distances than conventional vehicles would with the same energy consumption [105]. Adopting electric FAVs and managing road infrastructure can also reduce emissions and energy consumption.

### 6. Conclusions

We investigated the PV and WTB of FAVs and found that a higher PV toward FAVs is not necessarily correlated with a higher WTB, and vice versa. For those who are environmentally conscious, WTB would be high mainly because they believe that FAVs can resolve social problems such as air pollution and congestion. On the other hand, these people would not have higher PVs as they regard FAVs as not granting direct benefits; those who prioritize the conservation of the natural environment would not consider FAVs to increase, for example, biodiversity. Therefore, these people would not appreciate FAVs. Those who are afraid of possible accidents would not purchase FAVs and would present lower PVs. Using SEM, our model suggests that factors affecting WTB and PV are nonidentical.

Despite the time elapsed since our 2015 study and the ongoing development of FAVs, our findings remain valid and relevant. The early stage data offer foundational insights into consumer perceptions, crucial for understanding the evolution of public opinion towards such emerging technologies. The study's timing, prior to the widespread release of FAVs, actually enriches its significance, providing pre-market perspectives that can guide effective communication and development strategies. Furthermore, as FAV technology continues to advance, our study serves as a vital baseline for comparative analysis, offering a unique perspective on the dynamics of technology adoption and consumer interaction with transformative technologies.

Our study, while offering valuable insights, acknowledges several limitations and suggests areas for future research. Firstly, we employed SEM, which may involve concerns of reverse causality. For instance, individuals inclined to purchase FAVs might inherently per-

ceive their merits more positively and underplay potential accident risks. Addressing such reverse causality could be facilitated through an instrumental variable (IV) approach, using IVs that control for traits of early adopters or those with fixed demands. Unfortunately, our current dataset lacks variables that effectively differentiate these groups, presenting a limitation in our analysis.

Secondly, the global market has not yet seen the introduction of FAVs, a situation influenced more by legal considerations, particularly regarding accident liability, than technological readiness. Consequently, public perception of FAVs is based on a conceptual understanding rather than experience with the actual technology. This means that, while perceptions of FAVs might not significantly change over short periods, they could evolve over a longer span. Updating our data in future studies could enhance our understanding of these evolving perceptions and provide a more dynamic view of public attitudes towards FAVs.

In summary, our study's findings are constrained by methodological limitations and the current state of FAV technology and public perception. Future research could address these limitations by incorporating more sophisticated analytical approaches and updating data to capture the evolving landscape of FAV technology and its societal reception.

Another interesting future study is to employ discrete-choice methods. This study is interested in investigating the correlations of psychometric factors (which were expressed in latent variables) to the WTB and PV of FAVs; thus, we chose SEM. This study does not substantially discuss vehicle attributes such as fuel economy, weights, and sizes. Nonetheless, consumers may have some trade-off between vehicle attributes and automation functions. Applying discrete choice methods may capture trade-offs between different vehicle options, but it would require a completely different type of survey. For example, while this study investigates people's WTB and PV on 'adding' FAV options rather than purchasing an entirely new vehicle, discrete choice mostly requires buying data and information on the vehicle attribute. Most importantly, the discrete-choice model would require automobile prices, and our survey does not have price information, as we are not asking whether s/he is purchasing a new vehicle. Thus, discrete choice is left for future research.

**Author Contributions:** Conceptualization, J.K.; Methodology, Y.K.; Writing—original draft, S.Y. and A.R.K.; Supervision, S.M. All authors have read and agreed to the published version of the manuscript.

**Funding:** The authors would like to acknowledge the financial supports from Specially Promoted Research through a Grant-in-Aid (20H00648) from the Japanese Ministry of Education, Culture, Sports, Science and Technology (MEXT) and the Environment Research and Technology Development Fund (JPMEERF20201001) from the Japanese Ministry of the Environment.

**Informed Consent Statement:** Not applicable.

**Data Availability Statement:** The data presented in this study are available on request from the corresponding author. The data are not publicly available due to privacy.

**Conflicts of Interest:** The authors declare no conflict of interest.

**Appendix A**

Table A1 shows the proportion of respondents' evaluations of benefits and concerns regarding FAVs. We calculated the proportion as follows: the number of people who chose the option/sample size (N = 180,771).

Table A2 shows the proportion of respondents' evaluations on environmental awareness.

**Table A1.** Evaluation of benefits (latent construct: merit) and concern (Latent: fear).

| Latent Category 1: "Merit" | Evaluation (%) |
|---|---|
| People can drive without a license. | 28.78% |
| Burdens on driving would be decreased. | 66.27% |
| Children can move without a guardian. | 6.32% |
| Able to do other work while driving. (Multitask) | 45.12% |
| Able to avoid responsibilities of traffic accidents. | 32.89% |
| **Latent Category 2: "Fear"** | |
| There is a possibility that children will be able to move on their own. | 63.25% |
| There is a possibility that the software is hacked. (Cyber security) | 65.13% |
| The malfunction may cause accidents. | 80.23% |
| It is unclear who is responsible for the accident due to FAV technology. | 76.63% |
| Traffic volume and congestion might increase as those without a license can drive. | 52.98% |
| The malfunction may lead to wrong destinations. | 51.2% |

**Table A2.** Environmental awareness of respondents.

| Latent Category 3: "Pollution" | No Awareness | Very Important | Somewhat Important | Neither | Not Very Important | Not at All Important |
|---|---|---|---|---|---|---|
| Recycling is important. | 13.06% | 1.61% | 2.99% | 24.61% | 42.74% | 14.98% |
| Cycle utilization rate: the percentage of the total amount of reusable and recycled materials to be injected into society, is important for preventing pollution. | 13.50% | 1.83% | 3.49% | 27.30% | 41.40% | 12.48% |
| I think water quality should be improved. | 14.05% | 1.43% | 2.98% | 26.05% | 40.55% | 14.93% |
| Alleviating Particulate Matter (PM) 2.5. pollution is critical for our society. | 13.16% | 1.27% | 2.81% | 22.78% | 40.26% | 19.72% |
| Resolving air pollution (particularly, photochemical smog) is important. | 13.43% | 1.28% | 2.80% | 23.87% | 40.49% | 18.14% |
| **Latent Category 4: "Nature"** | | | | | | |
| Preserving endangered species is important. | 17.12% | 3.02% | 6.56% | 37.55% | 27.58% | 8.17% |
| Preserving living animals (overall) is important. | 16.22% | 3.87% | 9.64% | 41.24% | 23.05% | 5.98% |
| The ratio of green area within 1500 m around a house is important. | 15.09% | 2.41% | 6.09% | 35.36% | 32.96% | 8.09% |
| Green purchasing: When purchasing goods and services, consider the environmental impact before purchasing. | 15.40% | 2.66% | 5.60% | 38.95% | 29.51% | 7.89% |

Tables A3–A6 show the correlation matrix of indicator variables of latent constructs. Table A7 shows the results of the measurement equation, which describes the effects of latent constructs on each of indicator variables. Standardized coefficients are shown and all coefficients are positive and statistically significant at 0.001%. The values of the coefficients are almost unchanged across three specifications.

**Table A3.** Correlation matrix of the indicator variables of "Pollution".

|     | EP1 | EP2 | EP3 | EP4 | EP5 |
|-----|-----|-----|-----|-----|-----|
| EP1 | 1 | | | | |
| EP2 | 0.862 | 1 | | | |
| EP3 | 0.7957 | 0.8184 | 1 | | |
| EP4 | 0.7479 | 0.7503 | 0.8086 | 1 | |
| EP5 | 0.7531 | 0.7588 | 0.816 | 0.9035 | 1 |

**Table A4.** Correlation matrix of the indicator variables of "Nature".

|     | EN1 | EN2 | EN3 | EN4 |
|-----|-----|-----|-----|-----|
| EN1 | 1 | | | |
| EN2 | 0.8205 | 1 | | |
| EN3 | 0.7408 | 0.7357 | 1 | |
| EN4 | 0.6831 | 0.6766 | 0.6993 | 1 |

**Table A5.** Correlation matrix of the indicator variables of "Merit".

|     | MR1 | MR2 | MR3 | MR4 | MR5 |
|-----|-----|-----|-----|-----|-----|
| MR1 | 1 | | | | |
| MR2 | 0.2352 | 1 | | | |
| MR3 | 0.2954 | 0.1403 | 1 | | |
| MR4 | 0.0942 | 0.2794 | 0.131 | 1 | |
| MR5 | 0.3296 | 0.26 | 0.2284 | 0.1658 | 1 |

**Table A6.** Correlation matrix of the indicator variables of "Fear".

|     | FE1 | FE2 | FE3 | FE4 | FE5 | FE6 |
|-----|-----|-----|-----|-----|-----|-----|
| FE1 | 1 | | | | | |
| FE2 | 0.3162 | 1 | | | | |
| FE3 | 0.2317 | 0.3865 | 1 | | | |
| FE4 | 0.2362 | 0.3407 | 0.4249 | 1 | | |
| FE5 | 0.2944 | 0.3187 | 0.2771 | 0.3238 | 1 | |
| FE6 | 0.229 | 0.3531 | 0.3407 | 0.3257 | 0.3391 | 1 |

**Table A7.** Coefficient estimates of the measurement equation ($n$ = 180,771).

| Latent Variables | Model (1) | Model (2) | Model (3) |
|------------------|-----------|-----------|-----------|
| *Pollution* | | | |
| EP1 | 0.855 | 0.856 | 0.856 |
| EP2 | 0.872 | 0.873 | 0.873 |
| EP3 | 0.934 | 0.934 | 0.934 |
| EP4 | 0.867 | 0.867 | 0.867 |
| EP5 | 0.874 | 0.874 | 0.874 |

**Table A7.** *Cont.*

| Latent Variables | Model (1) | Model (2) | Model (3) |
|---|---|---|---|
| *Nature* | | | |
| EG1 | 0.910 | 0.910 | 0.910 |
| EG2 | 0.902 | 0.902 | 0.902 |
| EG3 | 0.816 | 0.815 | 0.816 |
| EG4 | 0.751 | 0.751 | 0.751 |
| *Merit* | | | |
| BD1 | 0.590 | 0.600 | 0.583 |
| BD2 | 0.483 | 0.476 | 0.490 |
| BD3 | 0.403 | 0.407 | 0.400 |
| BD4 | 0.433 | 0.428 | 0.437 |
| BD5 | 0.513 | 0.512 | 0.512 |
| *Fear* | | | |
| AC1 | 0.443 | 0.444 | 0.443 |
| AC2 | 0.618 | 0.619 | 0.618 |
| AC3 | 0.623 | 0.623 | 0.623 |
| AC4 | 0.607 | 0.607 | 0.607 |
| AC5 | 0.513 | 0.513 | 0.513 |
| AC6 | 0.546 | 0.546 | 0.546 |

Note: All coefficients are significant at $p < 0.001$.

Table A8 shows the distribution of the socio-economic variables of our sample and government statistics.

**Table A8.** Socio-economic distribution of the respondents and the Japanese population.

| | | Sample (%) ($n$ = 246,642) | Government Statistics (%) |
|---|---|---|---|
| Gender | Female | 41.0 | 51.3 |
| | Male | 59.0 | 48.7 |
| Education level | Junior high school or less | 2.1 | 9.5 |
| | High school | 26.9 | 42.3 |
| | Some college | 22.6 | 15.6 |
| | Bachelor/Master/Doctor | 45.9 | 23.9 |
| | Other | 1.9 | 8.6 |
| Age | 18–19 | 0.2 | 2.3 |
| | 20–29 | 5.4 | 11.7 |
| | 30–39 | 18.1 | 13.3 |
| | 40–49 | 31.9 | 17.2 |
| | 50–64 | 25.8 | 22.1 |
| | Over 65 | 10.7 | 33.4 |

**Table A8.** *Cont.*

|  |  | Sample (%)<br>(*n* = 246,642) | Government Statistics (%) |
|---|---|---|---|
| Household income | <2 million JPY | 7.8 | 18.3 |
|  | 2–3 million JPY | 8.9 | 17.2 |
|  | 3–4 million JPY | 11.9 | 15.3 |
|  | 4–5 million JPY | 12.3 | 12.2 |
|  | 5–6 million JPY | 11.9 | 9.0 |
|  | 6–7 million JPY | 9.6 | 6.9 |
|  | 7–8 million JPY | 9.1 | 5.8 |
|  | 8–9 million JPY | 6.9 | 4.1 |
|  | 9–10 million JPY | 6.7 | 3.4 |
|  | 10–15 million JPY | 10.5 | 6.0 |
|  | 15–20 million JPY | 2.7 | 1.1 |
|  | ≥20 million JPY | 1.7 | 0.7 |
|  | Do not know/do not want to answer | - | – |
| Region | Hokkaido | 4.6 | 4.2 |
|  | Tohoku | 5.9 | 6.9 |
|  | Kanto | 38.2 | 34.4 |
|  | Chubu | 16.6 | 16.8 |
|  | Kinki | 20.1 | 17.7 |
|  | Chugoku | 5.1 | 5.8 |
|  | Shikoku | 2.5 | 2.9 |
|  | Kyushu/Okinawa | 7.1 | 11.3 |
| Household size | 1 | 15.6 | 34.5 |
|  | 2 | 30.1 | 27.9 |
|  | 3 | 23.6 | 17.6 |
|  | 4 and above | 30.1 | 20.0 |

Sources: MIC (2013, 2015, 2017, 2019a, 2019b).

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
