# Peer review of "Buy and/or Pay Disparity: Evidence from Fully Autonomous Vehicles"

_applsci, doi:10.3390/app14010410_

Round 1

Reviewer 1 Report

Comments and Suggestions for Authors

The paper is a survey (very interesting, in fact), but the paper will be enhanced if the author will introduce the mathematical model used to analyze the data (e.g. the mathematical formulas).

1. What is the main question addressed by the research?
The main question (a very good one actually) is whether or not to buy an autonomous vehicle and at what price (the right one of course).

2. Do you consider the topic original or relevant in the field? Does it
address a specific gap in the field?
The topic is relevant.
Statistical models are used that should be explained (described) to be understood by someone who is not a specialist.
This would improve the quality of the work.

3. What does it add to the subject area compared with other published
material?
The statistical equation that described the proposed model.
It is not clear how the results have been obtained. They are raw figures (obtained from survey or this figures
have been processed (and how) to obtain the results?

4. What specific improvements should the authors consider regarding the
methodology? What further controls should be considered?
It is fine, but again a better explanation of Figure 2 should be made.

5. Are the conclusions consistent with the evidence and arguments presented
and do they address the main question posed?
The conclusion could be relevant if the previous observations will be clarified.

6. Are the references appropriate?
Yes.

7. Please include any additional comments on the tables and figures.
Table 4 - is incomplete or miss-aligned, please revisited.
Figure 2 must be explained in details (for example the meaning of the notations).

Author Response

RESPONSE TO REVIEWER 1

[Response to General Comments] Thank you for your thorough and positive review. We have revised our paper accordingly. The remainder of this letter contains your comments (italicized), our responses to each of them (written in black), and our revised points (highlighted in blue). Additionally, we have included the changes and additions to the paper that address your concernas for your reference.

[Comment 1] Table 4 - is incomplete or miss-aligned, please revisit.

[Our Response 1] Thank you for pointing out the issues with Table 4. We have thoroughly reviewed and revised this table to address the concerns of incompleteness and misalignment. In the revised version, we have ensured that:

  1. All entries are properly aligned for better readability and consistency.
  2. The notation, explanation, and source columns are clearly defined and uniformly formatted.
  3. Each category (Fear, Merits, Pollutants, Nature) and its respective items are grouped and presented in a structured manner.

We believe these revisions make the table more comprehensive and easier to interpret, thereby enhancing the quality of our manuscript. The updated version of Table 4 is included in the revised manuscript.

[Comment 2] Figure 2 must be explained in details (for example the meaning of the notations).

[Our Response 2] We appreciate your suggestion. To enhance readability, we have incorporated the following explanations on how to interpret each notation of the path diagram shown in Figure 2:

[Page 20] The rectangles in the diagram symbolize the observed variables, whereas the circles denote latent variables and error terms. Each arrow represents the path from one variable to another, with bidirectional arrows indicating correlations between variables. Each latent variable is measured by its corresponding indicator variables. Subsequently, four latent variables and individual characteristics have paths to our primary objective variables: WTB and PV. 

These revisions aim to clearly address your concerns by providing a more in-depth explanation of our data collection and analysis methods, thereby enhancing the clarity and robustness of our research methodology in the manuscript.

Reviewer 2 Report

Comments and Suggestions for Authors

In this paper a large-scale survey on FAVs of more than 180,000 respondents was collected in Japan to understand whether environment concerns, fear of potential accidents, and merits regarding fully autonomous vehicles (FAVs) are motivators of willingness to buy (WTB) and perceived value (PV) of FAVs. And use structural equation modeling (SEM) validated findings. The results show that those who would have purchased FAVs in the past if they were interested in natural environment conservation do not exhibit high PV. those who appreciate the potential benefits will have higher WTB and PV, while those who fear FAV technology will not purchase FAVs and will have lower PV. However, the following problems need to be solved:

1.The layout of some of the result maps needs to be adjusted to ensure that the effectiveness of the research results can be more clearly demonstrated.

2. The experimental data were comprehensive and whether multiple data processing methods were used for comparison.

3. The overall formatting of the article has a lot of typographical errors and needs to be adjusted in detail.

4. Table numbering and formatting are incorrect and need to be adjusted.

Comments on the Quality of English Language

 Moderate editing of English language required.

Author Response

 RESPONSE TO REVIEWER 2

[Response to General Comments] We sincerely appreciate your comprehensive and constructive feedback on our manuscript. In response, we have meticulously revised our paper to address the points raised. This letter delineates your comments, italicized for clarity, alongside our detailed responses, presented in standard black text. Furthermore, the amendments and enhancements made to the manuscript, in direct response to your observations. Enclosed herein are these revisions and additions, carefully crafted to address and incorporate your insightful suggestions, for your perusal and reference.

[Comment 1] The layout of some of the result maps needs to be adjusted to ensure that the effectiveness of the research results can be more clearly demonstrated.

[Our Response 1] Thank you for your feedback regarding the layout of our result maps. We have revised them to enhance clarity, readability, and accuracy. Key changes include improved color schemes, legible scaling and labeling, consistent formatting, and focused highlighting of crucial findings. These adjustments ensure a clearer demonstration of our research results in the revised manuscript.

[Comment 2] The experimental data were comprehensive and whether multiple data processing methods were used for comparison. 

[Our Response 2] We appreciate your inquiry regarding the comprehensiveness of our experimental data and the application of multiple data processing methods. In response to your comment, we have made the following amendments to our manuscript to better illustrate these aspects:

  1. Data Comprehensiveness: We have expanded the discussion in our methodology section to explicitly state the comprehensiveness of our survey data. This includes a detailed description of how the survey covered various demographic groups to represent the entire Japanese population accurately. We have now emphasized the large sample size of 246,642 respondents, which enhances the data's representativeness.

  1. Data Processing Methods: To address your point on data processing methods, we have added a new subsection that describes the various analytical techniques employed. This includes a comparative analysis with Japanese Census data, which is now more thoroughly detailed, particularly in Table A.5. We explain how this comparison helps validate the representativeness of our survey data despite minor discrepancies in gender and education levels.

These revisions aim to clearly address your concerns by providing a more in-depth explanation of our data collection and analysis methods, thereby enhancing the clarity and robustness of our research methodology in the manuscript.

[Comment 3] The overall formatting of the article has a lot of typographical errors and needs to be adjusted in detail.

[Our Response 3] Thank you for highlighting the typographical and formatting issues in our manuscript. We have thoroughly proofread the document to correct all typographical errors and adjusted the formatting for consistency with the journal's guidelines. These revisions enhance the manuscript's readability and professionalism.

[Comment 4] Table numbering and formatting are incorrect and need to be adjusted.

[Our Response 4] We appreciate your feedback on the table numbering and formatting. We have corrected the numbering of all tables and improved their formatting for clarity and consistency. These changes ensure accurate presentation and adherence to the journal's standards.

Reviewer 3 Report

Comments and Suggestions for Authors

The paper addresses an interesting topic with potential economic, social, and environmental impacts. Overall, I found the proposal quite intriguing, but I have some comments on the text.

Abstract - I suggest to the authors that in the abstract, the use of acronyms should be avoided.

I strongly advise the authors to review the formatting of the manuscript, which appears to be outside the standard required by the journal. Additionally, there are a number of tables whose boundaries extend beyond the text margins

Line 36 - Starting from line 36, the acronym FAV is written differently from how it has been previously described as 'F/AV'. I suggest to the authors to adopt a consistent form to harmonize the text.

Lines 138, 147, and 149 - What means "s/he". It is not understood.

I didn't understand why the research was conducted between November and December 2015, and only now in 2023 the work is being proposed. How can this time delay fail to capture the current scenario? I encourage the authors to provide a better explanation of this point in the manuscript.

Line 219 - I suggest converting the values from JPY to dollars

Table 2 is not referenced in the text

Table 3 - In some cases, the standard deviation is higher than the mean. Can the authors comment on these results?

I thought that the results were not explored in the best way. I suggest representing correlations through graphs, which can be more informative than the tables presented.

I missed a more detailed description of how the data was processed. I suggest that the authors revise the text to make this point clearer.

Comments on the Quality of English Language

Regarding the quality of the English language, I believe the text is well-written, needing only a few minor adjustments.

Author Response

 RESPONSE TO REVIEWER 3

[Response to General Comments] Thank you for your thorough and positive review. We have revised our paper accordingly. The remainder of this letter contains your comments (italicized), our responses to each of them (written in black), and our revised points (highlighted in blue). Additionally, we have included the changes and additions to the paper that address your concerns for your reference.

[Comment 1] Abstract - I suggest to the authors that in the abstract, the use of acronyms should be avoided. 

[Our Response 1] Thank you for your suggestion regarding the use of acronyms in the abstract. We have revised the abstract accordingly, removing all acronyms and replacing them with their full terms to ensure clarity and accessibility for all readers.

[Comment 2] I strongly advise the authors to review the formatting of the manuscript, which appears to be outside the standard required by the journal. Additionally, there are a number of tables whose boundaries extend beyond the text margins.

[Our Response 2] Thank you for your feedback on the manuscript's formatting and table layout. We have carefully revised the manuscript to align with the journal's formatting standards, including adjusting the size and alignment of all tables to ensure they fit within the text margins. These changes enhance the overall presentation and readability of the manuscript, ensuring compliance with the journal's requirements.

[Comment 3] Line 36 - Starting from line 36, the acronym FAV is written differently from how it has been previously described as 'F/AV'. I suggest to the authors to adopt a consistent form to harmonize the text.

[Our Response 3] Thank you for pointing out the inconsistency in the use of the acronym for Fully Autonomous Vehicles in our manuscript. We have revised the text to ensure that the acronym is consistently used throughout the document. We have chosen to uniformly adopt "FAV" for clarity and consistency, as per your suggestion. This amendment has been applied starting from line 36 and throughout the manuscript to maintain uniformity and avoid any confusion.

[Comment 4] Lines 138, 147, and 149 - What means "s/he". It is not understood.

[Our Response 4] Thank you for your observation regarding the use of "s/he" in our manuscript. We understand your concern about its clarity. The term "s/he" was employed as a gender-neutral pronoun, intended to inclusively represent "she or he." To ensure this is clearly understood by all readers, we have added a brief note in the manuscript's introduction to explain the use of "s/he" as an inclusive, gender-neutral pronoun. 

[Footnote 2] s/he" as an inclusive, gender-neutral pronoun, intended to inclusively represent "she or he."

[Comment 5]  I didn't understand why the research was conducted between November and December 2015, and only now in 2023 the work is being proposed. How can this time delay fail to capture the current scenario? I encourage the authors to provide a better explanation of this point in the manuscript.

[Our Response 5] We acknowledge the concern regarding the time gap between our study's conduct in November-December 2015 and its proposal in 2023, and its potential implications on capturing the current scenario surrounding Fully Autonomous Vehicles (FAVs). To address this, we include the following in the conclusion: 

[Revised Paragraph, Page 29] Despite the time elapsed since our 2015 study and the ongoing development of FAVs, our findings remain valid and relevant. The early-stage data offer foundational insights into consumer perceptions, crucial for understanding the evolution of public opinion towards such emerging technologies. The study's timing, prior to the widespread release of FAVs, actually enriches its significance, providing pre-market perspectives that can guide effective communication and development strategies. Furthermore, as FAV technology continues to advance, our study serves as a vital baseline for comparative analysis, offering a unique perspective on the dynamics of technology adoption and consumer interaction with transformative technologies.

In the manuscript, we have now included a detailed explanation of the study's timing and its relevance to the current state of FAV technology. This includes a discussion on the long-term value of our findings and how they contribute to the broader body of knowledge in the field of autonomous vehicles, irrespective of the specific technological advancements made since the data collection. 

[Comment 6]  Line 219 - I suggest converting the values from JPY to dollars 

[Our Response 6] Thank you for your suggestion to convert the values from JPY to USD in line 219 of our manuscript. We have updated the manuscript to reflect this change. Now, the range of perceived values (PV) for Fully Autonomous Vehicles (FAVs), originally stated as 0 to 3.25 million JPY, has been converted using the approximate rate of 1 JPY to 0.0073 USD. Accordingly, this range is now presented as approximately 0 to 23,725 USD, providing a clearer understanding for an international audience. This conversion enhances the manuscript's accessibility and relatability, especially for readers unfamiliar with JPY values. We have put a footnote. 

[Footnote 3] Based on the approximate conversion rate of 1 JPY to 0.0073 USD, the range of PV for FAVs from 0 JPY to 3.25 million JPY, converts to approximately 0 to 23,725 USD.​

[Comment 7] Table 2 is not referenced in the text 

[Our Response 7]  Thank you for highlighting the oversight regarding Table 2, which is currently Table 1, in our manuscript. We have reviewed the document and appropriately incorporated references to Table 1 within the text. This inclusion ensures that readers can easily connect the relevant discussions in the text to the data presented in Table 1, enhancing the coherence and flow of our argument. The revised manuscript now includes clear references to Table 1 in sections where its data and insights contribute significantly to our analysis and findings.

[Comment 8] Table 3 - In some cases, the standard deviation is higher than the mean. Can the authors comment on these results?

[Our Response 8] Thank you for your observation regarding Table 3, where in some cases, the standard deviation is higher than the mean. This statistical phenomenon can be attributed to several aspects of our data collection and the nature of the variables being measured. Firstly, the wide range of responses among participants leads to high variability, which is reflected in a greater standard deviation. This variation is indicative of the diverse opinions or experiences within the study group. Secondly, some variables in our data exhibit skewed distributions, where responses are clustered at one end of the scale. This clustering results in a low mean value but a high standard deviation due to the elongated tail of the distribution. Lastly, for variables with small mean values, particularly in data that are not normally distributed, it is common to observe a standard deviation that is larger than the mean. This is indicative of a broader spread of individual responses around a small central value. 

These statistical characteristics, inherent to our collected data, underscore the diversity of perceptions and experiences among the respondents, providing valuable insights into the range of opinions and attitudes within the study population. Importantly, we would like to emphasize that these aspects of data variability do not detract from the overall validity or integrity of our findings. Rather, they contribute to a more nuanced understanding of the data, enriching the conclusions and insights drawn from our study. The variability in the data adds depth to our analysis and is a valuable component of our research in understanding the dynamics of consumer perceptions and preferences.

[Comment 9] I thought that the results were not explored in the best way. I suggest representing correlations through graphs, which can be more informative than the tables presented.

[Our Response 9] Thank you for your suggestion to use graphical representations to illustrate the correlations in our results. We recognize the potential benefits of using graphs for visual clarity and ease of interpretation. However, due to the extensive number of variables in our study, creating graphs would either result in an overwhelming number of visualizations or necessitate the exclusion of some variables, which might compromise the comprehensiveness of our data presentation.

In response to your feedback, we have instead focused on revising and enhancing the tables presented in our manuscript. These revisions include improving the layout for better readability, employing clearer labeling, and optimizing the organization of data to facilitate easier comparison and interpretation. We believe these modifications effectively present the intricate relationships and patterns within our data, while maintaining the integrity and completeness of our findings.

We appreciate your recommendation and have strived to balance the clarity of presentation with the necessity to comprehensively represent the complex data set integral to our study's insights.

[Comment 10] I missed a more detailed description of how the data was processed. I suggest that the authors revise the text to make this point clear

[Our Response 10] We appreciate your inquiry regarding the comprehensiveness of our experimental data and the application of multiple data processing methods. In response to your comment, we have made the following amendments to our manuscript to better illustrate these aspects:

  1. Data Comprehensiveness: We have expanded the discussion in our methodology section to explicitly state the comprehensiveness of our survey data. This includes a detailed description of how the survey covered various demographic groups to represent the entire Japanese population accurately. We have now emphasized the large sample size of 246,642 respondents, which enhances the data's representativeness.

  1. Data Processing Methods: To address your point on data processing methods, we have added a new subsection that describes the various analytical techniques employed. This includes a comparative analysis with Japanese Census data, which is now more thoroughly detailed, particularly in Table A.5. We explain how this comparison helps validate the representativeness of our survey data despite minor discrepancies in gender and education levels.

These revisions aim to clearly address your concerns by providing a more in-depth explanation of our data collection and analysis methods, thereby enhancing the clarity and robustness of our research methodology in the manuscript.

Reviewer 4 Report

Comments and Suggestions for Authors

“Buy and/or Pay Disparity: Evidence from Fully Autonomous Vehicles” is a paper that makes policy recommendations based on a survey and includes many surveys and statistical analysis results.

However, although the current level is well presented, it is presented in the form of a report, so it is necessary to summarize the matters appropriate to this paper, and highlight the results of this study by extracting only the necessary matters.

It is necessary to check and revise the paper format as a whole.

The results and literature review are mixed, so the final intention of the paper is not clearly presented.

In this study, each question is described as a whole, so there are parts that make it less readable. It is necessary to edit the questions and answers into a table.

It would be good to describe the limitations of this study and future development directions in more detail.

Comments on the Quality of English Language

It would be good to make some changes to the language.

Author Response

RESPONSE TO REVIEWER 4

[Response to General Comments] Thank you for your thorough and positive review. We have revised our paper accordingly. The remainder of this letter contains your comments (italicized), our responses to each of them (written in black), and our revised points (highlighted in blue). Additionally, we have included the changes and additions to the paper that address your concerns for your reference.

[Comment 1] However, although the current level is well presented, it is presented in the form of a report, so it is necessary to summarize the matters appropriate to this paper, and highlight the results of this study by extracting only the necessary matters. 

[Comment 2] It is necessary to check and revise the paper format as a whole.

[Our Response to Comments 1 and 2]

In response to the insightful feedback from the reviewers, we have made significant revisions to our manuscript to enhance its presentation and relevance. Recognizing the need to transition from a report-style format to one more suited for an academic paper, we have carefully summarized and highlighted only the key aspects pertinent to our study. This includes a focused representation of our findings, ensuring that we extract and emphasize the most crucial elements of our research. For example, we have included the following which briefly summarizes our paper:

[Revised Paragraph, Page 3] Our results indicate that while environmental concerns enhance the likelihood of purchasing FAVs, they simultaneously diminish their perceived value due to the perception that FAV adoption does not directly benefit environmental conservation. On the other hand, individuals who recognize the benefits of FAV technology demonstrate a higher willingness to buy and assign greater value to these vehicles. Conversely, apprehensions regarding FAV technology correlate with a lower inclination to purchase and a reduced perceived value. These insights are pivotal in informing the development of effective policies and strategies for the adoption of FAVs.

Additionally, in line with the recommendations, we have overhauled the overall paper format for better clarity and academic rigor. Notably, Table 3 has been restructured to present the survey questions in a more organized and coherent manner compared to the previous version. These revisions aim to streamline the information and provide a clearer, more impactful presentation of our study's results, aligning with the standards expected in academic publications.

[Comment 3] The results and literature review are mixed, so the final intention of the paper is not clearly presented.

[Our Response 3] Thank you for pointing out the issue with the blending of results and literature review in our manuscript, which obscured the paper's final intention. In response to your feedback, we have revised the relevant sections to more clearly delineate and present the core objectives of our study. This includes an upfront statement of the paper's intention in the introduction, ensuring that readers are immediately aware of the study's goals and scope.

For instance, we have now explicitly outlined our primary aims at the beginning, followed by a structured literature review that supports these objectives without merging into the results section. This revision brings a clearer focus and structure to the manuscript, effectively separating the literature review from the results and highlighting the paper's main intentions and contributions.

We believe these changes address your concerns and enhance the overall coherence and clarity of the paper, making it easier for readers to understand the purpose and significance of our study.

[Revised Paragraph, page 4 ] The existing literature underscores the significance of examining the demographics adopting FAVs with a focus on the potential advantages and drawbacks of AVs or FAVs. This body of work posits that the anticipated benefits of AVs and FAVs can only be actualized through user acceptance and proper utilization. This entails avoiding pitfalls such as neglecting improvements in fuel efficiency or rapidly increasing travel distances due to overreliance on these vehicles, thereby highlighting the pivotal role of consumers. Informed by these insights, our research question is oriented towards exploring consumer behavior patterns, specifically in terms of choice and PV, in relation to AV/FAV adoption.

[Comment 4] In this study, each question is described as a whole, so there are parts that make it less readable. It is necessary to edit the questions and answers into a table.

[Our Response 4] Thank you for your valuable feedback on the readability of our manuscript concerning the presentation of survey questions and answers. In response to your suggestion, we have revised Table 3 to provide a clearer and more structured presentation of the questions and their corresponding answers.

This revision includes organizing the questions and answers into a tabular format, enhancing both readability and accessibility. The table now effectively condenses the information, allowing readers to easily comprehend and reference the survey content. This change addresses your concern about the narrative format's readability and ensures that the survey data is presented in a more user-friendly and concise manner. We believe that this modification significantly improves the clarity of our manuscript and aids in better understanding the structure and findings of our study.

[Comment 5] It would be good to describe the limitations of this study and future development directions in more detail.

[Our Response 5] Thank you for your suggestion to elaborate on the limitations of our study and the directions for future research. We have expanded our discussion in the manuscript to provide a more detailed account of these aspects.

[Revised Paragraph, Pages 29-30] Our study, while offering valuable insights, acknowledges several limitations and suggests areas for future research. Firstly, we employed SEM, which may involve concerns of reverse causality. For instance, an individual inclined to purchase FAVs might inherently perceive their merits more positively and underplay potential accident risks. Addressing such reverse causality could be facilitated through an instrumental variable (IV) approach, using IVs that control for traits of early adopters or those with fixed demands. Unfortunately, our current dataset lacks variables that effectively differentiate these groups, presenting a limitation in our analysis.

Secondly, the global market has not yet seen the introduction of FAVs, a situation influenced more by legal considerations, particularly regarding accident liability, than technological readiness. Consequently, public perception of FAVs is based on a conceptual understanding rather than experience with the actual technology. This means that, while perceptions of FAVs might not significantly change over short periods, they could evolve over a longer span. Updating our data in future studies could enhance our understanding of these evolving perceptions and provide a more dynamic view of public attitudes towards FAVs.

In summary, our study's findings are constrained by methodological limitations and the current state of FAV technology and public perception. Future research could address these limitations by incorporating more sophisticated analytical approaches and updating data to capture the evolving landscape of FAV technology and its societal reception.

Round 2

Reviewer 1 Report

Comments and Suggestions for Authors

In Table 4, please specify the means of "p" ( is a probability, a coefficient value?) and of values in the parenthesis. It is not clear (is a comparison with data in literature?) However, it is possibly I have missed that notation.

For example:
* p<0.1, ** p<0.05, *** p<0.01.
Nature -0.029***
(0.004)

Author Response

Thank you very much for your concern. For Table 4, we have included the following:

  • p<0.1, ** p<0.05, *** p<0.01. p stands for p-value. Standard errors in parentheses.

We believe this can help readers understand our research.

Reviewer 3 Report

Comments and Suggestions for Authors

The modifications made to the text, as well as the adjustments, have addressed my concerns. Therefore, I congratulate the authors on the work done.

Author Response

Thank you very much and we appreciate your time and efforts in this paper!

Reviewer 4 Report

Comments and Suggestions for Authors

I think it will be enough to finalize the format.

Author Response

(The authors gave the same response as above.)
